# Polymers Derived from Agro-Industrial Waste in the Development of Bioactive Films in Food

**DOI:** 10.3390/polym17030408

**Published:** 2025-02-04

**Authors:** Carlos Culqui-Arce, Diner Mori-Mestanza, Armstrong B. Fernández-Jeri, Robert J. Cruzalegui, Roberto Carlos Mori Zabarburú, Alex J. Vergara, Ilse S. Cayo-Colca, Juliana Guimarães da Silva, Nayara Macêdo Peixoto Araujo, Efraín M. Castro-Alayo, César R. Balcázar-Zumaeta

**Affiliations:** 1Instituto de Investigación, Innovación y Desarrollo para el Sector Agrario y Agroindustrial (IIDAA), Facultad de Ingeniería y Ciencias Agrarias, Universidad Nacional Toribio Rodríguez de Mendoza de Amazonas, Chachapoyas 01001, Peru; carlos.culqui@untrm.edu.pe (C.C.-A.); diner.mori@untrm.edu.pe (D.M.-M.); armstrong.fernandez@untrm.edu.pe (A.B.F.-J.); robert.cruzalegui@untrm.edu.pe (R.J.C.); roberto.mori@untrm.edu.pe (R.C.M.Z.); alex.vergara@untrm.edu.pe (A.J.V.); efrain.castro@untrm.edu.pe (E.M.C.-A.); 2Facultad de Ingeniería Zootecnista, Agronegocios y Biotecnología, Universidad Nacional Toribio Rodríguez de Mendoza de Amazonas, Chachapoyas 01001, Peru; icayo.fizab@untrm.edu.pe; 3Institute of Technology, School of Food Engineering, Federal University of Pará (UFPA), Belém 66075-110, PA, Brazil; prof.julianagsilva@gmail.com (J.G.d.S.); nayarapeixoto@ufpa.br (N.M.P.A.)

**Keywords:** natural polymers, sustainable packaging, antimicrobial, shelf life

## Abstract

This review explores the potential of biopolymers as sustainable alternatives to conventional plastics in food packaging. Biopolymers derived from plant or animal sources are crucial in extending food shelf life, minimizing degradation, and protecting against oxidative and microbial agents. Their physical and chemical properties, influenced by the raw materials used, determine their suitability for specific applications. Biopolymers have been successfully used in fruits, vegetables, meats, and dairy products, offering antimicrobial and antioxidant benefits. Consequently, they represent a functional and eco-friendly solution for the packaging industry, contributing to sustainability while maintaining product quality.

## 1. Introduction

Growing concern about the environmental impact of industries such as plastics and packaging has driven significant attention toward developing sustainable alternatives [1,2]. The excessive and often indiscriminate use of conventional plastics, particularly in food packaging, has accentuated critical ecological and sustainability challenges [3,4]. Derived primarily from fossil fuels, these plastics contribute significantly to global environmental issues, including the contamination of terrestrial and marine ecosystems and the emission of greenhouse gases during production and improper disposal. Such environmental consequences underscore the urgent need to transition toward eco-friendly and more sustainable materials [1,2].

Traditional plastics rely on non-renewable resources like petroleum, intensifying environmental concerns and raising questions about long-term resource availability. This has fostered an urgent call for alternative packaging materials that minimize ecological harm while addressing the increasing demand for sustainable practices. Furthermore, the concept of a circular economy is being highlighted, where materials are designed to be biodegradable or recyclable, thereby reducing the overall carbon footprint of packaging industries [5].

Packaging, as a crucial component of the food supply chain, serves vital functions in the food preservation chain due to its high protective capacity and ability to maintain quality during storage. It protects food products from physical, chemical, and biological damage while preserving their nutritional and sensory qualities [6,7]. However, current packaging solutions are often inadequate in addressing broader sustainability and food safety concerns. This limitation has spurred the exploration and adoption of biodegradable materials, such as edible films and coating, obtained from natural polymers of plant and animal origins [8]. It should be noted that edible biofilms and biocoatings are defined as a thin layer with the same manufacturing process to protect food from spoilage and as a carrier of flavoring, antimicrobial, and antioxidant agents [9,10,11].

Biodegradable packaging, particularly edible films and coatings, offers an innovative alternative to conventional plastics. These materials provide excellent barrier properties against spoilage and contamination while also serving as carriers for bioactive compounds, such as antimicrobials, antioxidants, and flavor enhancers [8,12,13,14]. Edible films, which are pre-formed and applied as protective layers, and edible coatings, which are directly applied to food surfaces, represent a dual approach to enhancing food safety and extending shelf life [9,10,11,15,16].

The production of these biodegradable materials relies on a diverse array of natural polymers. Polysaccharides, such as chitosan, gums, cellulose, and starch, along with proteins like gelatin, whey protein, egg, albumin, casein, and soy protein, are widely used [8,17]. Additionally, lipids are incorporated to improve water vapor barrier properties. Recent advances in material science have enabled the development of composite films that combine the strengths of various biopolymers, addressing limitations such as mechanical fragility and high moisture sensitivity [18,19].

Over the past years, research on edible films and coatings has intensified, with studies exploring single-component and composite formulations. This growing body of work has led to the emergence of sustainable, non-toxic, and easily degradable packaging solutions [9,20]. Market analysis projects that biopolymer-based packaging will experience an annual growth rate exceeding 14%, reflecting its alignment with global sustainability goals, such as those outlined in the 2030 Agenda for Sustainable Development [21].

The preparation and modification of edible films and coatings play a pivotal role in maximizing their potential as sustainable alternatives to conventional plastics. By tailoring their physicochemical properties through innovative techniques, these materials can meet the demands of modern packaging while contributing to global sustainability efforts [22]. Various methods, including casting, extrusion, and layer-by-layer assembly, are commonly employed to produce these films. Casting involves dissolving biopolymers in a solvent and drying to form a uniform film layer. Extrusion, on the other hand, applies heat and mechanical force to shape the film, allowing for large-scale production. Meanwhile, layer-by-layer assembly offers precise control over film thickness and composition by sequentially depositing layers of different biopolymers [23,24].

Several modification techniques have been explored to enhance the mechanical and barrier properties of edible films. These include crosslinking, which improves film strength and reduces water sensitivity, and the incorporation of plasticizers to increase flexibility. Additionally, blending biopolymers with nanoparticles, such as silver or clay, has shown promise in enhancing antimicrobial activity and improving the film’s overall structural integrity. Such advancements address the limitations of edible films and open new possibilities for their applications in diverse sectors beyond food packaging, including pharmaceuticals and cosmetics [25,26].

Therefore, given the increasing importance of these materials, this review seeks to analyze the various natural polymer sources used in developing biofilms and edible coatings. It also focuses on classifying these materials based on their origin, physicochemical properties, and practical applications within the food industry. Additionally, this review addresses key challenges, such as reduced mechanical strength, hygroscopicity, and limited barrier capabilities, while exploring potential solutions, including the integration of nanomaterials and multifunctional composites.

## 2. Natural Polymer Sources for Developing Bioactive Films

The food industry is replacing petroleum-based polymers, such as plastics, with biopolymers exhibiting different physicochemical characteristics, such as emulsification, gelation, stabilization, coagulation, film formation, and thickening [27,28]. Biodegradable polymers are formed by living organisms through enzyme-catalyzed and chain-growth reactions from monomers formed within cells [27,29]. Based on their origin, natural polymers are classified into two main categories (Figure 1).

### 2.1. Natural Polymers from Plant Sources

Cellulose is a natural biopolymer composed of D-glucose units linked by β-1,4-glycosidic bonds, serving as the primary structural component of plant cell walls [30]. Due to its abundance, biodegradability, and biocompatibility, cellulose has attracted significant attention as a sustainable alternative to petroleum-based plastics in applications such as food packaging and biodegradable films [31,32,33,34]. Its high molecular weight and strong fiber-forming tendency further enhances its potential for such applications.

The cellulose content varies depending on the species and age of the plant and can also be sourced from algae, animals, and bacteria [35]. Its structure, rich in hydroxyl groups (-OH), makes it highly reactive and ideal for chemical modifications, including esterification, crosslinking, and oxidation. However, cellulose films generally have low effectiveness as barriers to water vapor, primarily due to their hydrophilicity, which reduces their durability in humid environments [36]. To address this limitation, covalent functionalization processes are employed to improve their resistance, creating more effective barriers against water, oil, vapor, and oxygen.

In addition to plant-derived cellulose, bacterial cellulose offers unique advantages, such as high purity, mechanical strength, a crystalline structure, and excellent water retention capacity. These properties make it especially suitable for edible and eco-friendly packaging. When incorporated into food packaging films, bacterial cellulose provides high transparency, gloss, and effective barriers against gases and oil [37].

Lignin, the second-most abundant natural polymer after cellulose, forms a complex, amorphous 3D structure within plant cell walls. It consists of phenylpropanoid units (guaiacyl, syringyl, and p-hydroxyphenyl) linked by various ether and carbon–carbon bonds, providing resistance to degradation and offering the potential for high-value applications [38,39,40,41]. Due to its chemical composition, lignin exhibits properties such as high UV resistance, antioxidant activity, biocompatibility, and thermal stability [42].

Recent research highlights the use of lignin in composite films, such as starch- and PVA-based materials, which show significant improvements in barrier properties, UV resistance, and antibacterial functionality. Despite its advantages, the full use of lignin in industrial processes faces challenges related to its complex molecular structure and steric hindrances. This necessitates chemical modification strategies to expand its potential and foster the development of sustainable and technologically advanced materials [43].

Pectin, a heteropolysaccharide widely present in plant cell walls, is primarily extracted from industrial waste, such as citrus peels and apple pomace, which are considered sustainable resources due to their availability and high pectin content [44,45,46]. The degree of methoxylation influences the extraction process, which directly impacts its functional properties, such as gel formation and thickening capacity [45,47,48].

Beyond its food applications, pectin has been explored as a polymeric matrix for biodegradable packaging, thanks to its biodegradability, biocompatibility, and film-forming abilities, which provide suitable barrier properties to protect food from oxidation and microbial contamination [46]. The development of pectin-based composite films is a promising area, including incorporating bioactive agents and nanoparticles to enhance properties like hydrophobicity, mechanical resistance, and antimicrobial activity. Films enriched with tea polyphenols, titanium oxide, or essential oils have demonstrated increased efficacy in extending food shelf life [49]. Combining pectin with other polymers further improves thermal stability and barrier properties, underscoring its potential as a sustainable alternative to conventional plastics. These innovations align with the growing demand for environmentally responsible and safe food packaging solutions [44].

Starch is a carbohydrate widely used for films in food packaging and other applications due to its biodegradability, renewability, and ability to form transparent and flexible films. According to recent research, the authors incorporated microcrystalline cellulose (MCC) and N,N′-methylene diacrylamide (MBA) to form a dynamic network structure through dehydration–condensation reactions. This modification resulted in films with lower water absorption capacity, a higher contact angle, and improved mechanical strength, while maintaining transparency similar to pure starch films. The results indicated that the modified films have significant potential for packaging applications, offering a sustainable and functional alternative to conventional materials [50].

Zein, a plant protein derived from corn, exhibits promising characteristics for nanoparticle synthesis due to its biodegradability, biocompatibility, and ability to bind hydrophobic and hydrophilic compounds. These properties enable the encapsulation of various bioactive compounds, such as abamectin, lutein, insulin, quercetin, and essential oils, contributing to applications in controlled drug and nutrient delivery systems [51,52]. Zein’s capacity to form films and membranes positions it as a potential substitute for synthetic plastics, finding applications in food, pharmaceuticals, cosmetics, and coatings [45].

Processing methods, such as wet and dry milling or alkaline treatment, produce specific zein fractions (α, β, γ, and δ-zein) with distinct characteristics like solubility in hydroalcoholic media and molecular weights. Zein’s versatility, coupled with its renewable and biodegradable nature, makes it a sustainable alternative aligned with current environmental demands [53,54]. However, pure zein films exhibit mechanical limitations, such as brittleness, which has driven research into improving their properties by adding plasticizers, bioactives, and nanomaterials. These approaches broaden their applications, particularly in food preservation. For instance, zein coatings delay the ripening of fruits like tomatoes and mangoes and preserve peeled garlic by controlling weight loss, sugar content, and lipid oxidation [6,55].

Chemical modifications and combinations with nanoparticles, such as zinc and magnesium oxides, enhance the functionality of zein films, providing antimicrobial, UV-blocking, and water-resistance properties. These attributes position zein as a promising material for sustainable packaging and bioactive compound delivery systems, offering low water vapor permeability, excellent film-forming ability, and biocompatibility without directly competing with the food supply chain [53].

Lastly, gluten proteins, derived from plants, consist of monomeric gliadins and polymeric glutenins [56]. Protein-based films act as molecular barriers, controlling the transfer of moisture, oxygen, carbon dioxide, oils, fats, and volatile compounds, thereby preventing quality deterioration and extending food shelf life [57]. Additionally, their high strength, elasticity, and viscosity enable the production of tensile-resistant films [58]. The main sources are presented in Table 1.

### 2.2. Natural Polymers from Animal Sources

Collagen, a major structural protein in animal bones and skin, accounts for approximately 30% of total protein content. Collagen’s properties are classified into two main types: its gelation behavior, water retention, and film-forming properties [81], and its roles in protective colloidal, adhesion, cohesion, and emulsification [81,82] (Table 2).

Chitosan (CS), a natural polysaccharide derived from chitin found in the exoskeletons of crustaceans such as shrimp and crabs, stands out for its biocompatibility, biodegradability, safety, and antimicrobial properties [11,97]. Obtained through the deacetylation of chitin, its polycationic structure enables film formation, making it promising for food packaging and other applications, such as agriculture, medicine, and pharmaceuticals. However, pure chitosan films face challenges, including low mechanical strength and stability in aqueous solutions, which limit their industrial use. To overcome these limitations, blends with polymers like polyvinyl alcohol (PVA) and carboxymethyl cellulose (CMC), along with the addition of metal nanoparticles and natural active compounds, have been explored to create composite films with enhanced mechanical and functional properties [98,99].

The application of chitosan in nanoparticles (CSNPs) has also gained attention due to its excellent encapsulation and controlled release capabilities for bioactive compounds. Formed through ionic gelation between chitosan and sodium tripolyphosphate (TPP), these nanoparticles are widely used as carriers for essential oils and other bioactive compounds, improving their stability and bioavailability. In edible films, the incorporation of Pickering emulsions and potential biopolymers, such as polysaccharides and proteins, has shown promising results. These combined materials offer high antioxidant and antibacterial capacity, delaying the deterioration of packaged foods. For instance, tomatoes protected by chitosan composite films demonstrated better preservation than conventional plastics [99,100].

Keratin, a fibrous protein commonly found in feathers, hair, and horns, stands out as a sustainable biomaterial. Its composition includes amino acids such as glycine, valine, serine, and alanine, contributing to its diverse functionalities [101]. As a protein, keratin exhibits potential due to its inherent biocompatibility, biodegradability, and bioactivity [90]. With the growing interest in keratin-based biomaterials, keratin-based coatings, films, hydrogels, sponges, and mats have been extensively developed for applications in tissue engineering and regenerative medicine [102]. Keratin hydrolysates, obtained through enzymatic, chemical, or physical processes, offer benefits such as antioxidant, antimicrobial, and immunomodulatory activities, with applications in the food, cosmetic, and pharmaceutical industries [90,101]. Recent studies have combined keratin with polysaccharides like carboxymethyl cellulose to create packaging materials with superior UV barrier and responsive properties, offering innovative and eco-friendly solutions for food packaging [66].

Gelatin is a tasteless, colorless, and water-soluble protein produced from the partial or total hydrolysis of collagen, offering low cost and high availability [103]. Gelatin has been extensively studied for its film-forming ability, favorable functional properties, and usefulness as an outer barrier to protect foods from drying, oxygen exposure, and light [104]. Derived from various animal by-products, gelatin exhibits strong mechanical and barrier properties. Its triple-helix structure provides physical strength, while its amino acids readily absorb UV radiation, protecting packaged foods from oxidative damage [105]. In the food industry, gelatin’s functional properties, including water retention, film and foam formation, and emulsifying tendencies, make it a promising candidate for food preservation [106].

Gelatin-based films offer advantages such as robust water absorption, high reactivity of side chain groups, and excellent biosafety [107]. However, studies have noted that fish gelatin has a weaker structure, lower boiling point, and reduced amino acid content, resulting in poor mechanical strength. Adding chitosan has shown promise in addressing these shortcomings, significantly improving its performance [107,108,109].

## 3. Physical and Chemical Properties of Bioactive Films from Natural Polymers

It is crucial to assess the properties of food packaging films, which can be modified using various additives, such as crosslinking agents, plasticizers, texturing agents, antioxidants, and antimicrobial agents.

### 3.1. Mechanical Properties

Packaging films must exhibit excellent mechanical strength for food protection and commercialization. The mechanical properties of materials reflect their behavior when subjected to forces and heat [9]. Two key indicators—tensile strength (TS) and percentage elongation at break (%)—are used to evaluate these properties [110,111]. In this context, TS represents the maximum stress a film can endure before breaking, while the percentage elongation at break shows the maximum stretch of the film before rupture. Tensile strength is commonly measured following the American Society for Testing and Materials tests [112]. For example, research reported that adding epoxidized castor oil to soy protein-based films improved mechanical properties, increasing elongation at break by approximately 23% [113].

Lipids can enhance barrier and mechanical properties, but their concentration and nature may impact the film’s performance, emphasizing the need for careful proportioning. Excessive lipid addition or incompatibility with the matrix can lead to rigid, weak, and brittle films, reducing elongation, elastic modulus, and tensile strength [114]. Limited information exists on lipids’ role in enhancing biopolymers’ mechanical properties for food packaging. However, incorporating hydrophilic materials like proteins, starch, chitosan, and cellulose into lipid-based films can improve mechanical and thermal characteristics [115]. Furthermore, studies on starch-/lauric acid-based films found that ultrasound treatment, with amplitudes up to 40%, improved mechanical performance due to better lauric acid dispersion and amylose release from starch granule breakdown. This demonstrates that ultrasound can enhance the mechanical properties of starch–lipid composite films [116].

However, the mechanical property ranges for biopolymers used for food preservation can vary depending on the type of natural polymer used; for example, biofilms should have a minimum strength of 2–10 MPa and an elongation of 5–30% depending on the formulation to maintain the shape and protection of the food [117].

### 3.2. Water Vapor Permeability

The effectiveness of a film in maintaining food quality depends on its ability to regulate mass transfer between the internal structure of the food and its environment [118]. Barrier properties are crucial in determining the shelf life of food products [9] as they influence water vapor, gas migration, and the transfer of volatile organic compounds [119]. High-quality films with effective water vapor permeability prevent texture degradation, dehydration, shrinkage, and undesirable chemical reactions, thus preserving freshness [119]. A lower WVP value indicates a reduced rate of moisture transport through the packaging film, which increases moisture retention and minimizes water loss from the food [120].

Polysaccharide- and protein-based films, known for their relative humidity, good gas barrier properties, transparency, and gloss, generally exhibit low moisture retention. In contrast, lipid-based films offer excellent moisture retention but high oxygen permeability [121]. Controlling film viscosity is crucial to achieving the desired permeability, as lipid-based materials have extremely low water vapor permeability (WVP) values, making them effective moisture barriers due to their nonpolar characteristics [122]. However, the brittleness of lipids can cause adhesion and homogeneity issues when used alone, often necessitating combinations with hydrophilic hydrocolloids in emulsion or bilayer forms [123].

Studies show that adding waxes, such as candelilla, beeswax, and carnauba wax, to sodium caseinate, starch–gluten, and chitosan-based films significantly improves moisture barrier properties, an essential feature of quality food packaging [124]. It has been shown that emulsified films have the greatest influence on water vapor barriers, depending on the lipid type and polymer polarity [125]. It is also essential to consider that the barrier properties of biopolymer films can be affected by pores or cracks, facilitating water vapor diffusion. Ideal water vapor permeability ranges also depend on the type of food and the function the biofilm will perform. In general, food biofilms should have a water vapor permeability between 0.1 and 10 g-mm/m^2^-h (grams per millimeter of thickness per square meter per hour) [117].

### 3.3. Oxygen Permeability

Oxygen permeability measures the amount of oxygen that can pass through a film at specific temperature and relative humidity levels. Oxidative processes, such as lipid oxidation and enzymatic browning, can lead to degradation in foods like meats and affect and pigments, causing nutrient loss, off-flavors, discoloration, texture changes, and reduced shelf life [2]. Adding antioxidants, like citric acid, can help reduce oxidative reactions and spoilage.

Oxygen permeability is often evaluated through practical tests. For instance, Zhou et al. [126] examined the oxygen barrier efficiency of films by adding 3 mL of camellia oil to a test tube sealed with various films at 25 °C and 50% relative humidity. Results indicated that adding camellia oil improved the films’ oxygen permeability. Recent advancements in packaging materials tailored to consumer needs include bacterial nanocellulose membranes, which enhance water and oxygen barrier properties in polymer-based films, making them suitable for food packaging applications [127]. Oxygen barrier properties can also be enhanced through nanofiller incorporation [128]. For example, chitosan films embedded with exfoliated hexagonal boron nitride nanosheets (hBNNSs) demonstrated superior oxygen transmission rate barriers compared to pure films, with 96.35% reduction in oxygen permeability in the CS/hBNNS films [111]. This improvement occurs as nanofillers create a more complex pathway for oxygen molecules within the polymer, leading to better barrier properties in the nanocomposite film [129].

The oxygen permeability of food biofilms is also influenced by the type of product to be packaged and the storage conditions. For products requiring high protection against oxidation, such as fruit, fresh vegetables, meat, or dairy products, the oxygen permeability should be less than 1–10 cm^3^/m^2^-day-atm [130].

### 3.4. Antimicrobial Properties

Incorporating antimicrobial agents into films and coatings is essential for extending the shelf life of food products, which has generated considerable scientific and technological interest in recent years [9]. Edible films and coatings are treated with various antimicrobial agents that function in food through three mechanisms: agent release, immobilization, and absorption systems during storage, thereby enhancing food safety [131].

Notably, certain biopolymers possess inherent antimicrobial properties. For example, the amino groups in gelatin oligopeptide chains are believed to contribute to its antibacterial properties [2]. In chitosan, positively charged amino groups interact with negatively charged molecules on bacterial cell membranes, inhibiting growth and acting as an effective antimicrobial agent [132]. Additionally, research has examined various vegetable oils and animal fats, including corn, sunflower, olive oil, and butter, to assess their impact on the environmental properties of chitosan-based packaging materials. The study found that olive oil and chitosan films exhibited the highest antimicrobial properties, comparable to the commercial antibiotic gentamicin [133].

Waxes such as carnauba, beeswax, and candelilla have also been used in developing bioactive coatings [9]. These studies highlight the potential of non-starch polysaccharides, particularly chitosan, for creating active materials in food packaging. Future research should investigate whether other non-starch polysaccharides can also provide antioxidant and antimicrobial activities in films.

### 3.5. Antioxidant Properties

Chemical degradation in food products occurs due to prolonged exposure to air, heat, or light, leading to the generation of reactive oxygen species [2]. To address this, packaging materials with antioxidant properties are being developed by incorporating active compounds into the polymeric matrix to preserve the quality of fruits and vegetables [134]. Examples of lipid-based antioxidants used in food packaging include essential oils, phospholipids, phenolipids, and lipophenols. Phenolic compounds and flavonoids in these components contribute to their antioxidant properties [114].

Lignin, a natural antioxidant, has been shown to enhance the antioxidant activity of k-carrageenan films, though only up to a particular concentration [135]. Other studies have demonstrated that sodium caseinate films containing ginger and cinnamon essential oils and sunflower protein concentrate films with clove essential oil in varied concentrations exhibit good antioxidant properties [9]. Additionally, beeswax has antioxidants, antimicrobial, healing, anti-stress, and anti-inflammatory properties, while carnauba wax has antioxidant and antiprotozoal properties [136].

The range of antioxidant activity varies depending on the type of biopolymer, the incorporation of bioactive compounds and additives, and external factors (pH, temperature, and interaction with microorganisms). This property is based on preserving foods to extend their shelf life and maintain their organoleptic quality [117].

### 3.6. Thermal Stability

The thermal stability of packaging materials is crucial as it affects their durability during storage, transport, handling, and disposal through incineration [137]. Thermal stability is evaluated using thermogravimetric analysis, which measures the mass loss of films as the temperature increases; greater mass loss indicates lower thermal stability [138]. Researchers have used non-starch polysaccharides to enhance the thermal stability of starch-based films, such as those made with Mesona chinensis and chitosan [139,140]. Non-starch polysaccharides can reduce water loss from composite films during heating, improving thermal stability for certain applications [141].

Incorporating carbon nanotubes has been shown to increase the thermal stability of hydroxypropyl starch films, mainly due to an increase in film crystallinity [142]. For thermoplastic banana peel films, initial weight loss at 150 °C was attributed to free water evaporation, while significant mass loss at 290 °C was due to starch degradation [143]. Adding nanocellulose fibers from banana peel improved thermal stability, raising the degradation temperature by 15 °C.

The required thermal stability ranges may vary depending on the type of biopolymer used and the specific application. For example, food packaging requires a glass transition temperature (Tg) above 40–60 °C so that the biofilm does not lose its structure during storage at room temperature [144].

### 3.7. Biodegradability

Biodegradability is a property of films used in food packaging [145]. The biodegradation of polymers is a complex natural process that is difficult to fully replicate in a laboratory setting due to various biogeochemical factors. Consequently, testing polymer biodegradation currently offers limited flexibility [146].

In a biodegradation test, Chinese chive root extract was added to the carboxymethyl cellulose film. The highest extract concentration (5% *w*/*w*) showed the greatest weight loss—58.14% after three weeks—compared to 32.66% in the control film [97]. This improvement in biodegradability is attributed to the formation of new polymeric materials and reduced environment degradation loads. Similarly, soil degradability tests on chitosan films containing rosemary and sage extracts showed that rosemary extract resulted in approximately 99% degradability over five days, while sage extract films degraded by 95% [147]. At the time of disposal, the biofilm must be able to biodegrade under humidity and heat conditions, and microorganisms in soil or composting systems. This means it must decompose at industrial composting temperatures (50 and 60 °C) and in the natural environment [148].

### 3.8. Optical Properties

The optical characteristics of food films, including color, opacity, and light transmission, influence consumer acceptance [149]. Packaging properties depend on the type and nature of additives used [150]. Color, a fundamental element, affects the overall appearance of a food product and strongly impacts consumer decisions [99]. Studies have shown that adding tea extract can significantly change packaging color, as tea polyphenols contain chlorophyll, carotenoids, and other yellow-green pigments [151].

Packaging transparency and light transmittance are also critical, especially when films are applied to the surface of foods [152]. These properties influence consumer choice and play a role in protecting against light, preventing transmission (particularly UV light), and inhibiting lipid oxidation, discoloration, and nutrient loss [153,154]. Because UV light can trigger food oxidation, using packaging films with strong UV-blocking properties is essential to prevent discoloration and nutritional loss from photooxidation [155,156,157]. The optical properties of food-grade biofilms are critical because of their visual appearance, with a preferred crystallinity index of 42–56% [158].

### 3.9. Solubility

Solubility is a fundamental characteristic of biofilms that directly impacts their applications in food systems. Many polysaccharides, such as cellulose, chitosan, alginate, and starch, are being used in food packaging, but their solubility limits their usefulness, and chemical or physical modifications are often required to adjust this property, such as changing the pH. For example, the high crystallinity of cellulose makes it insoluble in polar solvents, restricting its use in certain production techniques [98,159]. To address this, derivatives such as carboxymethyl cellulose with improved solubility and thermoplastic behavior have been developed [98,160]. Similarly, starch is insoluble in cold water but undergoes partial solubilization and loss of crystallinity upon heating, influencing the properties of starch-based films [98]. Innovations such as the conjugation of polysaccharides with catechins have shown promise in increasing solubility. For example, inulin–catechin conjugates demonstrate superior solubility in several solvents compared to inulin [161], while starch dialdehyde conjugated with catechins exhibits improved water solubility due to polysaccharide degradation during the reaction [162]. These advances expand the applicability of polysaccharides in the food industry, particularly in sustainable packaging solutions. Studies reported that biofilms must have less than 30% solubility in water, depending on the type of biopolymer and its formulation, preventing in this way the film from dissolving or disintegrating rapidly when it comes into contact with moisture or food liquids [163].

### 3.10. Viscosity

The viscosity of polysaccharides is a critical factor in producing biopolymer films. Research suggests polysaccharide–catechin conjugates with optimal viscosity are suitable for creating food packaging films, edible coatings, and hydrogels for the food industry [164]. However, few studies have investigated the viscosity of polysaccharide–catechin conjugates [165,166]. Some researchers found that arabinoxylan–catechin conjugates obtained via a free radical-mediated reaction had lower viscosity than arabinoxylan alone, due to the degradation effect of free radicals on polysaccharide chains [165]. Consequently, polysaccharide–catechin conjugates produced by acidic condensation reactions may also show reduced viscosity. In another study, Kim et al. [166] synthesized chitosan–catechin conjugates through a catalyzed reaction, finding a 3% increase in chitosan viscosity after conjugation with catechins, possibly due to the increased molecular weight of the conjugate.

Research has demonstrated that the viscosity of polysaccharide–catechin conjugates is influenced by the degree of substitution and the specific polysaccharide used [165,166]. Future studies should investigate the viscosity of polysaccharide–catechin conjugates produced by an acidic condensation reaction, as well as the impact of the synthesis method and type of catechin on the viscosity of these conjugates [164]. Finally, studies report the viscosity of biofilms between from 1 Pa·s to as high as 10^8^ Pa·s [167].

### 3.11. Emulsification Properties

The emulsification property of biopolymers refers to their capacity to stabilize mixtures of two immiscible liquids, such as oil and water. This characteristic is important in diverse applications, from food to pharmaceuticals and cosmetics. The natural water affinity of polysaccharides and the hydrophobicity of certain polyphenols offer an interesting approach to endow polysaccharides with emulsifying properties [168]. Given the relevance of polysaccharide and polyphenol conjugates in biopolymer films, Q. Li et al. [169] studied three distinct conjugates isolated from Camellia sinensis (brick tea), finding that each conjugate exhibited antioxidant and emulsifying properties. A recent study examined potato protein and quercetin conjugates as Pickering emulsion nano-stabilizers [170].

Similarly, the conjugation increased the protein’s particle size, resulting in larger emulsion droplets. Additionally, introducing hydrophobic polyphenols through conjugation can enhance the surface hydrophobicity of proteins, thereby improving emulsifying properties [171]. Studies report that the criteria for determining an emulsifier’s emulsion-stabilizing capacity is its ability to maintain at least 50% of the original emulsion volume [172].

## 4. Application of Biopolymer Films in the Food Industry

During storage, foods undergo changes in pH and gaseous composition, along with the production of spoilage compounds, such as amines, ammonia, and hydrogen sulfide, leading to decay [173]. Biopolymer-based films thus serve as an effective alternative for food packaging [174]. Biopolymers are increasingly reinforced with nanoparticles, and functional compounds like antibacterial, antioxidant, texturizing, and emulsifying agents are often incorporated [119]. Various biopolymers are currently used to extend the shelf life of certain foods (Figure 2).

### 4.1. Fresh Fruits and Vegetables

Fruits and vegetables are highly perishable during postharvest processes, including harvesting, storage, and transportation (Figure 3). Edible coatings based on cellulose nanoparticles and lemongrass essential oil have been developed the extend blackberry shelf life by up to six days in storage [175]. Similarly, a coating of 0.5% pectin and nanocellulose extended the shelf life of *Physalis Peruviana* fruits relative to a control coating [176]. A study compared two coatings—one made of chitosan and Myrtus communis essential oil and another with chitosan–essential and nanocellulose—for strawberry preservation, finding that the coating with nanocellulose extended the strawberries’ shelf life by up to 24 days [177].

Moreover, Oyom et al. [178] developed an edible coating based on modified sweet potato starch and cumin essential oil for pear preservation, which maintained quality over 28 days at 25 °C. The coating delayed color change, reduced decay, slowed respiration, and minimized weight loss. A gelatin-based coating with cranberry juice was also used to maintain the nutraceutical quality of tomatoes, with cranberry juice’s bioactive compounds enhancing the coating’s antimicrobial activity [179]. Another study found that coating raisins with a monoglycerol stearate and polysorbate solution preserved their sensory and microbiological quality for 12 weeks [180].

### 4.2. Meat Products

Fish, meat, and their derived products spoil rapidly if not properly preserved. Jiang et al. [181] wrapped fresh pork in a collagen-based film (Figure 3) derived from herbivorous carp, encapsulated lemon essential oil with chitosan (GCC/CS-LEO). The film effectively extended the pork shelf life to 21 days at 4 °C, whereas unwrapped pork was deemed unacceptable by day 9. Aitboulahsen et al. [182] studied tilapia fish filets using fish skin gelatin films with pectin, Mentha pulegium essential oil (MEO), and Lavandula angustifolia essential oil (LAEO), observing a positive effect on bacterial growth inhibiting over 11 days at 4 °C.

Similarly, Khorshidi et al. [183] investigated the chemical and microbial properties of vacuum-packed chicken leg coated with chitosan enriched with essential oils and Elletaria cardamomum extracts, extending shelf life to 12 days, compared to 2–3 days with chitosan-only films. Duan et al. [184] evaluated chicken meat freshness using films made from chitosan, gelatin, starch, raspberry anthocyanin (RA), and curcumin, which displayed excellent antibacterial and antioxidant properties. Additionally, Wang et al. [185] used a chitosan film enriched with apricot seed essential oil (AKEO) for seasoned beef packaging, observing antimicrobial properties against L. monocytogenes and improved texture, flavor, color, and acceptability. Qiu et al. [186] developed a nanoemulsion-based coating of chicken bone gelatin (CBG) and chitosan for ready-to-eat chicken burgers, extending their shelf life by 4 days and reducing moisture loss during storage. By day 8, uncoated burgers had fully degraded, while those with nanoemulsion showed lower pH, bacterial count, and volatile basic nitrogen at 4 °C. Similarly, Mohammadi et al. [187] developed gluten-based edible films with beeswax and diacetyl tartaric acid ester monoglycerides, reducing staling and improving hamburger bun quality, resulting in reduced hardness and rancidity.

### 4.3. Dairy Products

Similarly, plant-based protein coatings have been applied to cheeses, reducing oxidation [188]. Wrapping ricotta cheese in a fish gelatin film containing orange peel pectin can also extend its shelf life [189]. Brown et al. [190] coated fresh cheese with chitosan and chitosan, both with and without various antimicrobials, finding that chitosan combined with 5% hydrogen peroxide was more effective against L. monocytogenes than chitosan alone. In Kesar cheese storage, whey protein isolate (WPI) films containing oregano, garlic essential oil, nisin, and natamycin inhibited microbial activity. Seydim et al. [191] reported a 3-log reduction in E. coli counts on WPI film samples with oregano oil and added nisin by the end of storage, while control samples had 8.22 log CFU/cm^2^. All WPI films with essential oils also inhibited the growth of L. monocytogenes and S. enteritidis. Similarly, soft cheese was packed in WPI film with clove oil (CO) and peppermint oil (PO) at different concentrations, remaining free of microbial contamination for 30 days. Arshad et al. [192] observed the highest DPPH scavenging activity of 84.96% on day one of storage in WPI films with 1% added CO, compared to 76.80% in control samples. Although antioxidant activity gradually decreased over time, films with higher CO concentrations retained a greater scavenging capacity (66%) at the end of the storage period.

## 5. Future Perspectives

The need for biodegradable alternatives for food preservation drives the demand for edible films and coatings. With the growing market for edible packaging, it is estimated that the global value of edible films and coatings for fruits and vegetables will reach 4.79 billion USD by 2029, with an annual growth rate of 7.64% between 2024 and 2029. Edible coatings offer benefits such as enhancing antioxidants, antimicrobial, and barrier properties for packaging materials (Figure 4). However, they present challenges, particularly regarding the compatibility and solubility of biopolymers, which can lead to uneven distribution within the packaging material. Therefore, research on biopolymers from animal and plant origin is crucial for large-scale industrial applications. Future research should focus on aspects such as the bioavailability of biofilms in food, controlled release systems, their interaction with the gut microbiome, and microbial safety studies—areas that could provide innovative solutions for food preservation and safety.

Additionally, studying the effects of biofilms on the human body is essential, particularly their role in chronic infections, their influence on the immune system, and their potential relationship with antimicrobial resistance. Optimizing biofilm production protocols is also important, focusing on tailoring formulations for different foods, incorporating bioactive compounds, and applying advanced techniques like electrospinning and 3D printing. By using renewable raw materials, the environmental impact of these packaging options can be reduced, promoting sustainability in the food industry.

While biopolymer-based materials represent a sustainable alternative to synthetic polymers, their functionality is still limited compared to conventional plastics. Currently, no single biopolymer possesses all desired properties, such as moisture resistance, film-forming ability, mechanical and thermal properties, and colorlessness, odorlessness, and transparency. Blending different biopolymers is a promising strategy to enhance the functionality of these materials. In summary, the challenge lies in selecting synergistic ingredients that optimize the properties of edible packaging, using techniques such as plasticizers and gelling agents. Although research has made significant progress, there are still hurdles to practical application in the food industry. Animal- and plant-based biopolymers are well positioned as sustainable and functional alternatives for the future of food packaging.

## 6. Conclusions

Biopolymers are ideal materials for packaging applications. Materials intended for this purpose must possess specific properties, such as gas permeability (e.g., oxygen), water vapor permeability, tensile strength, elasticity, and thermal stability. Although biopolymers have superior characteristics, it is possible to enhance their mechanical and oxygen barrier properties for even greater performance. This can be achieved through various modification techniques, such as conjugation, crosslinking, and reinforcement with nanofillers, making them more suitable for industrial uses. These properties can be further refined by using additives like plasticizers, texturizers, antioxidants, and antimicrobial agents.

Edible coatings and films share similar functions and composition; both aim to improve food preservation by serving as barriers to humidity, gases, and microbial contamination. Also, they can reduce environmental impact because biopolymers can be derived from agro-industrial waste. In addition, edible coatings are beneficial for products like fruits and vegetables, while films are often used for more robust packaging needs. Therefore, both can be used in various food preservation scenarios on broad preservation strategies.

This review also highlights several biopolymers of animal and plant origin, such as chitosan, gelatin, collagen, keratin, casein, whey, starch, and pectin. Additionally, it addresses the recent development of biopolymers from these sources as packaging materials. Life cycle analysis indicates that these biopolymers have a lower environmental impact due to their natural origin, making them safe for various applications. Therefore, animal- and plant-based biopolymers offer distinct properties and applications, positioning them as potential sustainable packaging material alternatives for the future.

## Figures and Tables

**Figure 1 polymers-17-00408-f001:**
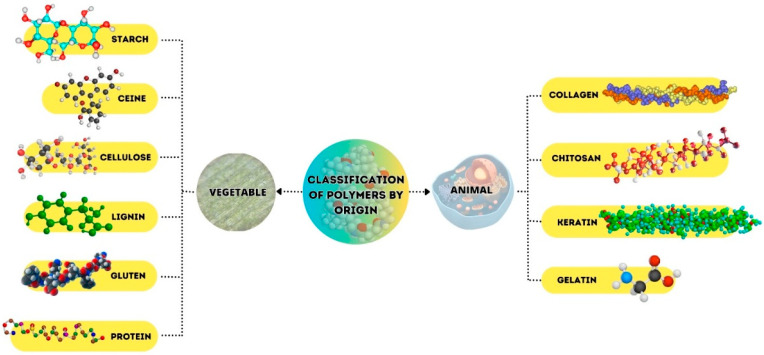
Polymers obtained from natural sources.

**Figure 2 polymers-17-00408-f002:**
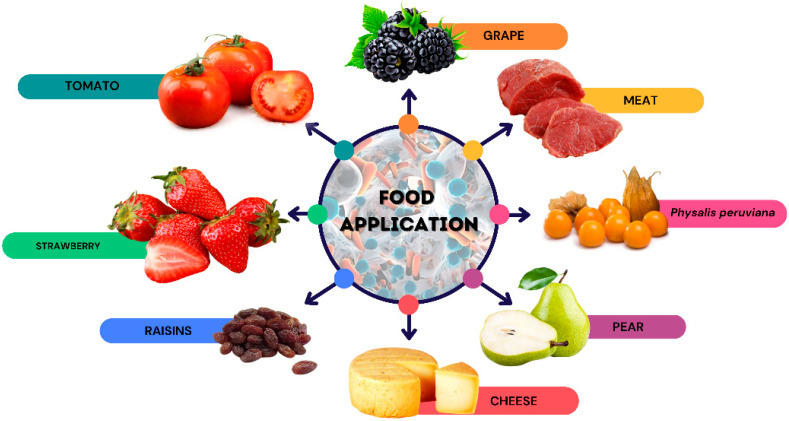
Application of natural polymers in food.

**Figure 3 polymers-17-00408-f003:**
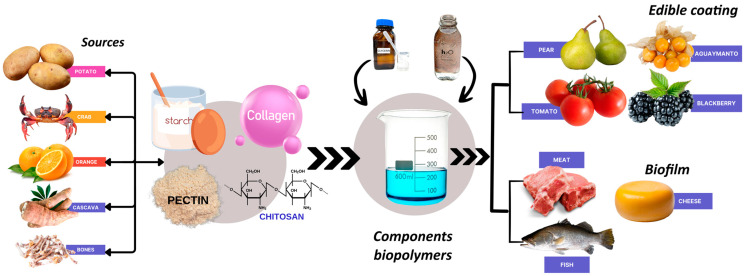
Biopolymer solutions in fresh fruits and vegetables, meat, and dairy products.

**Figure 4 polymers-17-00408-f004:**
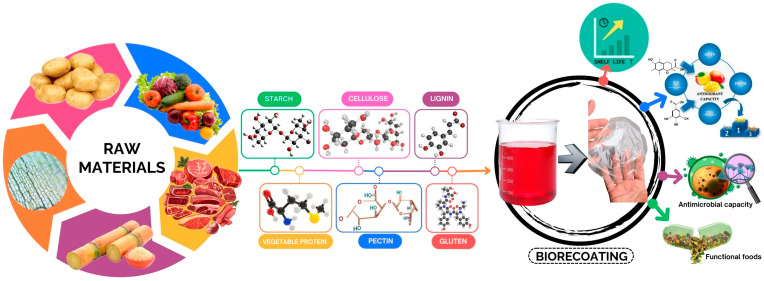
Trends in the use of edible coatings on foods.

**Table 1 polymers-17-00408-t001:** Main polymers reported from plant sources(↑: High ↓: Low).

Polymers	Source of Obtaining	Advantages	Disadvantages	References
Starch	Tubers, banana, coix and lotus seeds, and kudzu	↑ Water retention capacity and retention rate; ↓ absorption temperature; and strengthens the immune system	↓ Mechanical resistance; ↑ sensitivity to humidity; and ↓ barrier protection	[59,60,61,62]
Cellulose	Cell walls of wood, plants, algae, and bagasse	↑ Molecular weight; ability to form crystalline fibers; ability to retain water; and water solubility and gelling properties	↓ Flexibility; ↑ sensitivity to humidity; and ↓thermal stability	[35,63,64]
Lignin	Wood, ground walnut shells, raw rice straw, sugar cane bagasse, and corn stalk	↑ Antibacterial property; and ↑ carbon content and thermal stability	↓ Mechanical resistance; ↓ barrier protection; and raw materials are scarce	[65,66,67,68,69]
Vegetable proteins	Leaves, roots, and stems (alfalfa, radish, spinach, bamboo, beet, spinach, tomato, cabbage, and broccoli)	Source of essential amino acids; and ↑ water and oil absorption capacity	↓ Mechanical resistance; ↑ hydrosensitivity; and ↓ barrier protection	[12,70,71,72]
Pectin	Citrus, sugar beet, apple, and grapefruit peel	Thickening and gelling agent; anti-inflammatory, anticancer, reduces cholesterol and intestinal microbiota, and stimulates the immune response	↓ Mechanical resistance; ↑ hydrosensitivity; ↓ barrier protection; ↓ thermal stability, and sensitivity to enzymes	[73,74,75,76]
Zein	Corn	Ability to encapsulate compounds; and ↓ immunogenicity properties	↑ Hydrosensitivity; ↓ barrier protection; ↓ flexibility; and ↓ mechanical resistance	[52,53,77,78]
Gluten	Wheat	Solubility, foaming properties, and emulsification	↑ Sensitive to humidity; ↓ barrier protection; ↓ thermal stability; and ↓ mechanical resistance	[57,79,80]

**Table 2 polymers-17-00408-t002:** Main polymers reported from animal sources(↑: High ↓: Low).

Polymers	Source of Obtaining	Advantages	Disadvantages	References
Collagen	Skin, bones, and connective tissues	Film formation; self-assembly and gel formation	↑ Degradation and stability; ↓ mechanical resistance; and may generate allergic reactions	[83,84,85,86]
Chitosan	Shells of insects, fish, and vertebrates	Biodegradable, biocompatible, film formation, antimicrobial, polycationic, non-allergic, anticholesteric agent, moisturizing agent, absorption, renewable, soluble, viscosity, and swelling	↑ Sensitivity to humidity; ↓ mechanical resistance; possible toxic effects; and ↑ sensitivity to pH	[2,86,87,88,89]
Keratin	Horns, claws and nails, wool, and feathers	Inherent biocompatible, biodegradable, bioactive, and thermally stable; and ↓ amino acid content	↓ Mechanic resistance; ↑ sensitivity to environmental factors; and processing complexity	[86,90,91]
Gelatin	Pork skin and bones, cuttlefish and fish, and calfskin	↑ Film forming capacity, strength, nutritional value, and functional properties, and can be used as an edible film	↑ Sensitivity to humidity, ↓ mechanical resistance, and ↓ thermal stability	[92,93,94,95,96]

## Data Availability

The data are available in the article.

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
