# Peer review of "Polymers Derived from Agro-Industrial Waste in the Development of Bioactive Films in Food"

_polymers, 2025, doi:10.3390/polym17030408_

Round 1

Reviewer 1 Report

Comments and Suggestions for Authors

Dear authors

The manuscript is about biopolymers applying in food packaging. The paper is useful and interesting for readers. Please consider following comments:

1-In all tables, it is better to write advantages and disadvantages of every biopolymer instead of total properties.

2-Please add a schematic figure for various applications of mentioned biopolymers at the end of related text.

3-In section 3, please add some examples for required properties in special food industry application. For example, please write required properties in special application as a number range. Please do it for all mentioned properties composed of mechanical, thermal, viscosity, and so on.

Sincerely

Author Response

Thank you for your comments. We addressed every suggestion to improve the manuscript, and below are the modifications made:

 (1) In all tables, it is better to write advantages and disadvantages of every biopolymer instead of total properties.

Reply: Thank for the suggestion. We wrote advantages and disadvantages in all tables (L. 196-Table 1, and L. 202-Table 2)

(2) Please add a schematic figure for various applications of mentioned biopolymers at the end of related text.

Reply: Thank for the comment. We added the figure 3 in the main text [L. 502-503].

(3) In section 3, please add some examples for required properties in special food industry application. For example, please write required properties in special application as a number range. Please do it for all mentioned properties composed of mechanical, thermal, viscosity, and so on.

Reply: Thank you for the suggestions. We added information [L. 281-284, 310-313, 334-337, 373-376, 392-395, 408-411, 424-426, 443-446, 464-465, and 479-481]

Reviewer 2 Report

Comments and Suggestions for Authors

In this paper, this review discusses biopolymers as a substitute for plastics in food packaging, highlighting them as an essential alternative for promoting sustainability. Overall, this manuscript was well written and high quality. I recommend it publish on Foods before major revision.

1. Abstract should be rewritten, make it clear.

2. Introduction is too short.

3. Natural Polymer Sources: Pectin is also from natural polymer.

4.The author has introduced too many properties of films, but not the preparation and modification methods of films.

5. Please cite this reference about starch film:

  • DOI: 10.1016/j.ijbiomac.2024.129404

6. Why the authors present the edible coatings? This review is about film.

Comments on the Quality of English Language

No

Author Response

Thank you for your comments. We addressed every suggestion to improve the manuscript, and below are the modifications made:

(1) Abstract should be rewritten, make it clear.

Reply: Thank for the suggestion. We rewrote the abstract [L. 18-26]

(2) Introduction is too short.

Reply: Thank you for the comment. We explained and added information in the introduction [L. 30-100]

(3) Natural Polymer Sources: Pectin is also from natural polymer.

Reply: Correct, pectin is also derived from natural polymers. We explained information about it [L. 142-157]

(4) The author has introduced too many properties of films, but not the preparation and modification methods of films.

Reply: Thank for the suggestion. We added the information (L. 76-93)

(5) Please cite this reference about starch film: DOI: 10.1016/j.ijbiomac.2024.129404

Reply: Thank for the support. We considered the reference about starch film (L. 158-166).

(6) Why the authors present the edible coatings? This review is about film.

Reply: Thank you for the comment. We explained about edible coatings and film [L. 53-55, 598-604]

Round 2

Reviewer 2 Report

Comments and Suggestions for Authors

Accept

Comments on the Quality of English Language

No